# Semi-Analytical Analysis of Drug Diffusion through a Thin Membrane Using the Differential Quadrature Method

**Abdelfattah Mustafa** [1], **Reda S. Salama** [2,*] and **Mokhtar Mohamed** [2]

1   Department of Mathematics, Faculty of Science, Islamic University of Madinah, Madinah 42351, Saudi Arabia; aelsayed@iu.edu.sa or amelsayed@mans.edu.eg
2   Basic Science Department, Faculty of Engineering, Delta University for Science and Technology, Gamasa 11152, Egypt; mokhtar.alsaidi@deltauniv.edu.eg
*   Correspondence: reda.salama@deltauniv.edu.eg or prof.dr.reda.salama@gmail.com; Tel.: +20-1061391656

**Abstract:** The primary goal of this work is to solve the problem of drug diffusion through a thin membrane using a differential quadrature approach with drastically different shape functions, such as Lagrange interpolation and discrete singular convolution (the delta Lagrange kernel and the regularized Shannon kernel). A nonlinear partial differential equation with two time- and space-dependent variables governs the system. To reduce the two independent variables by one, the partial differential equation is transformed into an ordinary differential equation using a one-parameter group transformation. With the aid of the iterative technique, the differential quadrature methods change this equation into an algebraic equation. Then, using a MATLAB program, a code is created that solves this equation for each shape function. To ensure the validity, efficiency, and accuracy of the developed techniques, the computed results are compared to previous numerical and analytical solutions. In addition, the L∞ error is applied. As a consequence of the numerical outcomes, the differential quadrature method, which is primarily based on a discrete singular convolution shape function, is an effective numerical method that can be used to solve the problem of drug diffusion through a thin membrane, guaranteeing a higher accuracy, faster convergence, and greater reliability than other techniques.

**Keywords:** group theoretic method; drug diffusion; differential quadrature technique; discrete singular convolution; thin membrane; Lagrange interpolation polynomial

**MSC:** 34A25

## 1. Introduction

Diffusion through membranes represents a fascinating form of diffusion research that is crucial in the pharmaceutical industry [1–4]. To study the procedure, multiple experiments can be carried out. Testing a drug's non-stationary diffusion across a donor cell through a thin membrane to a recipient cell is one experiment that clearly shows this behavior [2,3]. The problem of drug diffusion through a thin membrane was examined in 2002 utilizing the group theoretical technique, which is considered a special case of the Lie method and yielded limited results. This method was used to determine the drug concentration in the membrane and that in the cells of the donor and receiver [4]. Spoelstra and Van Wyk [5] modeled the process as a pair of cells of identical volume divided by a thin membrane called cells of the donor and the receiver. A high drug concentration dissolves in a saltwater solution in the cell of a donor, while the receiver's cell possesses only a solution of saline. Since both cells are subjected to continual stirring, the drug does not initially diffuse through the membrane. The concentration of the drug starts to rise in the receiver cell and across the membrane. The authors used the finite difference technique to solve the problem numerically, evaluating the results of the model's parameters and determining the concentration in receiving and donor cells. It was demonstrated that

diffusion through membranes can be modeled using specific boundary conditions across them at various scales, from macroscopic to microscopic [6]. In this case, the membranes are considered thin in relation to the overall size of the system. The membrane is introduced as a transmission boundary condition on a macroscopic scale, allowing effective modeling of systems involving multiple scales. A numerical lattice Boltzmann scheme with a partial bounce-back condition at the membrane was proposed and analyzed on a microscopic scale [7]. This microscopic approach was shown to provide an accurate approximation of the transmission boundary condition. Moreover, a formula for the permeability of a thin membrane as a function of a microscopic transmission parameter was derived from the analysis of the macroscopic scheme. In a microscopic model, the mean waiting time for a particle to pass through the membrane is proportional to the membrane's permeability [8–10].

The macroscopic transmission conditions were discretized by Aho et al. [11] in order to derive an expression for the mass flux between two discretization points separated by a membrane. The expressions used to solve the diffusion equation were obtained from the finite difference and finite volume methods. The authors proposed a microscopic implementation of the lattice Boltzmann method [12,13], in which the membrane is treated as a partial bounce-back condition between two lattice nodes. They performed a multiple-scale analysis after presenting the discrete scheme and derived an explicit expression for the mass flux at the membrane. Several papers in the literature have presented mathematical models that use the group transformation technique, such as Hansen [14], Gaggioli et al. [15], Boutros et al. [16], and Abd-el-Malek et al. [17], to calculate the diffusion of a drug through a membrane along with that in both donor and receiver cells. Abd-el-Malek et al. [18] recently used the Lie group method to investigate drug diffusion through a biological membrane that typically partially absorbed the drug. The mathematical scheme was defined by a nonlinear partial differential formula which included the process of diffusion along the initial and boundary conditions and afterwards was converted into an ordinary differential equation with its associated conditions by reducing the number of independent variables by one. Fourth- and fifth-order Runge–Kutta methods were used to solve the obtained nonlinear ordinary differential equation.

Finite difference, finite element, finite volume, meshless, and least squares methods have been used to solve such problems. The main disadvantage of these methods is the requirement of a high number of grid points, such that a long performance time is required to obtain the required accuracy. Recently, a differential quadrature method (DQM) has become the most popular method for deriving numerical solutions to boundary value problems. It was developed by Richard Bellman and his associates in the early 1970s [19,20]. The key idea of the DQ application lies in the calculation of the weighting coefficients for the first-order derivative based on the Lagrange polynomial function. This method leads to accurate solutions with fewer grid points compared with finite difference and finite element methods. The convergence and stability of this method depend on the choice of the shape function. Lagrange interpolation polynomials, the cardinal sine function, the delta Lagrange kernel (DLK), and the regularized Shannon kernel (RSK) are some examples of such functions which have led to the development of the polynomial-based differential quadrature method (PDQM) [21], the sinc differential quadrature method (SDQM) [22], and the discrete singular convolution differential quadrature method (DSCDQM), respectively [23]. Thus, the DQ method has emerged as a powerful numerical discretization tool for solving a variety of problems in the engineering and physical sciences [24–30]. Bellman et al. [19] suggested that the nth-order derivative of the function with respect to a grid point can be approximated as a linear summation of the values of the function for all of the sample points in the domain.

In our model, we use a differential quadrature method (DQM) with drastically different shape functions to solve the problem of drug diffusion through a thin membrane. These shape functions are the Lagrange interpolation function [21,31], the Delta Lagrange kernel, and the Regularized Shannon kernel [32–35], and they have been successfully applied to the

problem of drug diffusion through a thin membrane. Furthermore, the partial differential equation is transformed into an ordinary differential equation (ODE) using a one-parameter group transformation [4] to reduce the two independent variables by one. To ensure the validity, efficiency, and accuracy of the developed techniques, the computed results are compared to previous numerical and analytical solutions [4,18]. The novelty of this method is to provide a parametric analysis in two cases to explain how time and the membrane section affect the drug concentration and diffusion coefficient for the first time.

## 2. Formulation of the Problem

Fick's-law-based diffusion mathematical models do not apply to human skin, hair-free rodent skin membranes, or multiple synthetic membranes [5]. Fick's laws of diffusion describe diffusion and were derived by Adolf Fick in 1855. They can be used to calculate the diffusion coefficient. Fick's first law can be used to derive his second law, which in turn is identical to the diffusion equation. Assuming that the rate at which absorption occurs at any place via the membrane corresponds to the drug concentration at that place and that no diffusion occurs through the membrane's edges, the concentration gradient through the membrane is viewed as a function of time and x direction, and the velocity of diffusion in the membrane for each unit area is related to the concentration gradient through the membrane. Consider a unit thickness membrane, wgere "q" is the coefficient that calculates the rate of absorption of the drug in the membrane if "Q" is the partition coefficient. The governing equation of the diffusion process is [4]:

$$\frac{\partial C(x,t)}{\partial t} = \frac{\partial}{\partial x}\left[P(x,t)\frac{\partial C(x,t)}{\partial x}\right] - q\left[\frac{C(x,t)}{t}\right]^2, \ 0 < x < 1, \ 0 < t \tag{1}$$

where $C(x,t)$ is the drug's concentration at an instant of time $(t)$ and at an x-distance from a single side of the membrane. The function utilized to compute the diffusion coefficient is denoted by $P(x,t)$.

Based on the boundary condition (Equations (2) and (3)) and initial conditions (Equation (4)):

$$\begin{cases} C(0,t) = F(t) \\ C(1,t) = G(t) \end{cases} \quad t > 0 \tag{2}$$

$$F(t) = \frac{\alpha Q}{D(t) - \beta}, \ \ G(t) = \gamma Q[R(t)]^2 \tag{3}$$

$$C(x,0) = 0 \ \ \text{at } 0 \le x \le 1 \tag{4}$$

where $\alpha$, $\beta$, and $\gamma$ are constants. In addition, $D(t)$ and $R(t)$ are the concentrations of drug in the donor and the receiver cells, respectively.

## 3. Method of Solution

We use a differential quadrature approach with totally different shape functions to solve a nonlinear partial differential equation with two dependent variables, $C(x,t)$ and $P(x,t)$, in both space and time. To begin, we solve the partial differential Equation (1) using a one-parameter group transformation. In this transformation, the two independent variables are minimized by one, and this equation is transformed into an ODE shown in the following:

Assuming the identity function, i.e., $\zeta = x$, according to a similar analysis, the dependent variables "C" and "P" are

$$C(x,t) = \psi(t)U(\zeta) \tag{5}$$

$$P(x,t) = \omega(t)T(\zeta) \tag{6}$$

Thus, using Equations (5) and (6), this is accomplished by substituting "C" and "P" and their partial derivatives as follows:

$$\begin{cases} \frac{\partial C}{\partial t} = U \frac{d\psi}{dt} \\[2mm] \frac{\partial C}{\partial x} = \psi \frac{dU}{d\zeta} \\[2mm] \frac{\partial^2 C}{\partial x^2} = \psi \frac{d^2 U}{d\zeta^2} \\[2mm] \frac{\partial P}{\partial x} = \omega \frac{dT}{d\zeta} \end{cases} \tag{7}$$

Then, by substituting Equation (7) into (1):

$$U(\zeta) \frac{d\psi(t)}{dt} = \frac{\partial}{\partial \zeta}\left[\omega(t)T(\zeta)\psi(t)\frac{dU(\zeta)}{d\zeta}\right] - q\left[\frac{\psi(t)U(\zeta)}{t}\right]^2 \tag{8}$$

By simplifying Equation (8)

$$\frac{\partial^2 U}{\partial \zeta^2} + \left[\frac{1}{T}\frac{\partial T}{\partial \zeta}\right]\frac{\partial U}{\partial \zeta} - \left[\frac{1}{\omega\psi T}\frac{\partial \psi}{\partial t}\right]U - q\left[\frac{\psi}{\omega T t^2}\right]U^2 = 0 \tag{9}$$

To reduce Equation (9) to an expression with one independent invariant ($\zeta$), the coefficients must be constants or functions of ($\zeta$) only. Thus,

$$\begin{cases} \frac{1}{T}\frac{\partial T}{\partial \zeta} = E_1(\zeta) \\[2mm] \frac{1}{\omega\psi T}\frac{\partial \psi}{\partial t} = E_2(\zeta) \\[2mm] \frac{\psi}{\omega T t^2} = E_3(\zeta) \end{cases} \tag{10}$$

*3.1. Case 1*

Assume $T(\zeta) = exp(-\zeta)$, $\psi(t) = \mu\, t$, and $\omega(t) = \frac{1}{t}$, where $\mu$ is a constant, Hence, $E_1 = -1$, $E_2 = exp(\zeta)$, and $E_3 = \mu\, exp(\zeta)$. So, Equation (9) can be written as:

$$\frac{\partial^2 U}{\partial \zeta^2} - \frac{\partial U}{\partial \zeta} - exp(\zeta)U - q\mu\, exp(\zeta)U^2 = 0 \tag{11}$$

with the following boundary conditions:

$$\begin{cases} U(0) = \alpha Q \\[2mm] U(1) = Q \end{cases} \tag{12}$$

In addition,

$$\begin{cases} P(x,t) = \frac{exp(-x)}{t}, \\[2mm] D(t) = \frac{1}{\mu t} + \beta, \\[2mm] R(t) = \sqrt{\frac{\mu}{\gamma}t}. \end{cases} \tag{13}$$

*3.2. Case 2*

Assume $T(\zeta) = exp(-\zeta^2)$, $\psi(t) = \mu\, t$, and $\omega(t) = \frac{1}{t}$, where $\mu$ is a constant. Hence, $E_1 = -2$, $E_2 = exp(\zeta^2)$, and $E_3 = \mu\, exp(\zeta^2)$. Thus, Equation (9) can be written as:

$$\frac{\partial^2 U}{\partial \zeta^2} - 2\zeta\frac{\partial U}{\partial \zeta} - exp\left(\zeta^2\right)U - q\mu\, exp(\zeta^2)U^2 = 0 \tag{14}$$

with the following boundary conditions:

$$\begin{cases} U(0) = \alpha Q \\ U(1) = Q \end{cases} \tag{15}$$

In addition,

$$P(x, t) = \frac{\exp(-x^2)}{t} \tag{16}$$

Now, for the previous cases, the differential quadrature method (DQM) with various shaped functions is employed to solve nonlinear partial differential equations as follows:

- Shape Function 1: Lagrange Interpolation Polynomial (PDQM);

The functional values associated with any unknown such "U" at a specific number of grid points "N" can be defined by this shape function as [21,31,36]:

$$U(\zeta_i) = \sum_{j=1}^{N} \frac{\prod\limits_{k=1}^{N} (\zeta_i - \zeta_k)}{(\zeta_i - \zeta_j) \prod\limits_{j=1, j \neq k}^{N} (\zeta_j - \zeta_k)} U(\zeta_j), \ (i = 1 : N) \tag{17}$$

As a result, the following are the various derivatives of "U":

$$\left. \frac{\partial^n U}{\partial \zeta^n} \right|_{\zeta = \zeta_i} = \sum_{j=1}^{N} \Re_{ij}^{(n)} U(\zeta_j), \quad (i = 1 : N) \tag{18}$$

where $\Re_{ij}^{(n)}$ denotes the weighting coefficients for the nth derivative of "U".

As a result, we need the weighting coefficients for the first and second derivatives ($\Re_{ij}^{(1)}$ and $\Re_{ij}^{(2)}$) to solve these cases, which can be discovered by differentiating Equation (17):

$$\Re_{ij}^{(1)} = \begin{cases} \dfrac{1}{(\zeta_i - \zeta_j)} \prod\limits_{\substack{k = 1, \\ k \neq i, j}}^{N} \dfrac{(\zeta_i - \zeta_k)}{(\zeta_j - \zeta_k)} & i \neq j \\ -\sum\limits_{\substack{j = 1, \\ j \neq i}}^{N} \Re_{ij}^{(1)} & i = j \end{cases}, \ \Re_{ij}^{(2)} = \left[ \Re_{ij}^{(1)} \right] \left[ \Re_{ij}^{(1)} \right], \tag{19}$$

- Shape Function 2: Discrete Singular Convolution (DSCDQM)

The singular convolution is presented as [34,37–39]:

$$g(\zeta) = (W * \eta)(\zeta) = \int_{-\infty}^{\infty} W(\zeta - s) \, \eta(s) \, ds \tag{20}$$

where $W(\zeta - s)$ denotes a singular kernel and $\eta(\zeta)$ is a function space element for testing.

This shape function is determined by the kernel type. However, because this shape function has many kernels, we will use two of them to describe the functional values of "U" and its derivatives at a given number of grid points "N" as follows:

Kernel (1): Delta Lagrange Kernel (DLK):

The DSC typically uses a weighted linear sum of the function values at 2M+1 points in the direction of the space variable to approximate the derivative of a certain function

with regard to a space variable at a discrete point [31,32,40]. The DLK can be applied as a shape function to approximate unknown "U" and its derivatives as the following:

$$U(\zeta_i) = \sum_{j=-M}^{M} \frac{1}{(\zeta_i - \zeta_j)} \times \frac{\prod\limits_{k=-M}^{M} (\zeta_i - \zeta_k)}{\prod\limits_{j=-M,\, k \neq i,j}^{M} (\zeta_j - \zeta_k)} \times U(\zeta_j), \quad (i = -N : N), \ M \geq 1 \qquad (21)$$

As a result, the following are the various derivatives of "U":

$$\left.\frac{\partial^n U}{\partial \zeta^n}\right|_{\zeta = \zeta_i} = \sum_{j=1}^{N} \Re_{ij}^{(n)} U(\zeta_j), \quad (i = -N : N) \qquad (22)$$

As a result, we need the weighting coefficients for the first and second derivatives ($\Re_{ij}^{(1)}$ and $\Re_{ij}^{(2)}$) to solve these cases, which can be discovered by differentiating Equation (21):

$$
\begin{aligned}
\Re_{ij}^{(1)} &= \begin{cases} \dfrac{1}{(\zeta_i - \zeta_j)} \prod\limits_{\substack{k=-M,\\ k \neq i,j}}^{M} \dfrac{(\zeta_i - \zeta_k)}{(\zeta_j - \zeta_k)} & i \neq j \\[2em] -\sum\limits_{\substack{j=-M,\\ j \neq i}}^{M} \Re_{ij}^{(1)} & i = j \end{cases}\!\!, \\[3em]
\Re_{ij}^{(2)} &= \begin{cases} 2\left( \Re_{ij}^{(1)} \Re_{ii}^{(1)} - \dfrac{\Re_{ij}^{(1)}}{(\zeta_i - \zeta_j)} \right) & i \neq j \\[2em] -\sum\limits_{\substack{j=-M,\\ j \neq i}}^{M} \Re_{ij}^{(2)} & i = j \end{cases}\!\!,
\end{aligned}
\qquad (23)
$$

Kernel (2): Regularized Shannon kernel (RSK) [41]:

To make comparisons and demonstrations, the regularized Shannon kernel is used to discuss this problem. In DSCDQM–RSK, it is presumed that the unknown "U" with its derivatives is the estimated weighted linear sum of the nodal values. As a result, the regularized Shannon kernel is discretized by [41]:

$$U(\zeta_i) = \sum_{j=-M}^{M} \left\langle \frac{\sin\left[\frac{\pi(\zeta_i - \zeta_j)}{\Delta}\right]}{\frac{\pi(\zeta_i - \zeta_j)}{\Delta}} \exp\left(\frac{-(\zeta_i - \zeta_j)^2}{2\sigma^2}\right) \right\rangle U(\zeta_j) \qquad (24)$$

where $(i = -N : N), \sigma = (\tau \times \Delta) > 0$

where $\Delta, \tau$, and $\sigma$ are the step size, the computational parameter, and the factor regularized Shannon respectively. The truncation error is very tiny due to the use of the Gaussian regularizer; thus, the above version provided by Equation (24) is feasible and has basically compact numerical interpolation support.

As a result, the following are the various derivatives of "U":

$$\left.\frac{\partial^n U}{\partial \zeta^n}\right|_{\zeta = \zeta_i} = \sum_{j=1}^{N} \Re_{ij}^{(n)} U(\zeta_j) \quad (i = -N : N) \qquad (25)$$

As a result, we need the weighting coefficients for the first and second derivatives ($\mathfrak{R}_{ij}^{(1)}$ and $\mathfrak{R}_{ij}^{(2)}$) to solve these cases, which can be discovered by differentiating Equation (24):

$$
\begin{aligned}
\mathfrak{R}_{ij}^{(1)} &= \begin{cases} \dfrac{(-1)^{i-j}}{\Delta(i-j)}\exp(-\Delta^2(\frac{(i-j)^2}{2\sigma^2})), & i \neq j \\[2mm] 0 & i = j \end{cases}, \\[3mm]
\mathfrak{R}_{ij}^{(2)} &= \begin{cases} \left(\dfrac{2(-1)^{i-j+1}}{\Delta^2(i-j)^2}+\dfrac{1}{\sigma^2}\right)\exp\left(-\Delta^2\left(\dfrac{i-j}{\sqrt{2}\sigma}\right)^2\right), & i \neq j \\[2mm] -\dfrac{1}{\sigma^2}-\dfrac{\pi^2}{3\Delta^2} & i = j \end{cases}
\end{aligned}
\tag{26}
$$

This discussion has revealed that the kernel type, regular grid points (N), and bandwidth (2M+1) play a significant role in achieving convergence and accuracy solutions.

The linear ODE is then obtained using the iterative quadrature method, which is a numerical method used to solve initial value problems (IVPs) for ordinary differential equations (ODEs). This method involves the use of numerical quadrature formulas to approximate a solution of the ODE, and then iteratively improving the approximation until the desired level of accuracy is achieved.

1.　Firstly, solving Equations (11) and (13) as a linear system;

Case 1:

$$
\sum_{j=1}^{N}\mathfrak{R}_{ij}^{(2)}U_j - \sum_{j=1}^{N}\mathfrak{R}_{ij}^{(1)}U_j - \exp(\zeta)\sum_{j=1}^{N}\delta_{ij}U_j - q\mu\exp(\zeta)\sum_{j=1}^{N}\delta_{ij}U_j = 0
\tag{27}
$$

Case 2:

$$
\sum_{j=1}^{N}\mathfrak{R}_{ij}^{(2)}U_j - 2\zeta\sum_{j=1}^{N}\mathfrak{R}_{ij}^{(1)}U_j - \exp\left(\zeta^2\right)\sum_{j=1}^{N}\delta_{ij}U_j - q\mu\exp(\zeta^2)\sum_{j=1}^{N}\delta_{ij}U_j = 0
\tag{28}
$$

2.　Then, we solve the following iterative system until the required convergence is reached [31,40];

$$
\left|\frac{U_{s+1}}{U_s}\right| < 1
$$

where $s = 0, 1, 2, \ldots$

Case 1:

$$
\sum_{j=1}^{N}\mathfrak{R}_{ij}^{(2)}U_{s+1,j} - \sum_{j=1}^{N}\mathfrak{R}_{ij}^{(1)}U_{s+1,j} - \exp(\zeta)\sum_{j=1}^{N}\delta_{ij}U_{s+1,j} - q\mu\exp(\zeta)\times \\ \left[\sum_{j=1}^{N}\delta_{ij}U_{s,j}U_{s+1,j}\right] = 0
\tag{29}
$$

Case 2:

$$
\sum_{j=1}^{N}\mathfrak{R}_{ij}^{(2)}U_{s+1,j} - 2\zeta\sum_{j=1}^{N}\mathfrak{R}_{ij}^{(1)}U_{s+1,j} - \exp\left(\zeta^2\right)\sum_{j=1}^{N}\delta_{ij}U_{s+1,j} - q\mu\exp(\zeta^2)\times \\ \left[\sum_{j=1}^{N}\delta_{ij}U_{s,j}U_{s+1,j}\right] = 0
\tag{30}
$$

So, the key to DQM accuracy is finding the weighting coefficients, which are based on the appropriate selection of a shape function.

### 4. Numerical Results

In this section, the DQM is examined with totally different shape functions (PDQM [21,31,36], DSCDQM–DLK, and DSCDQM–RSK [34,37–39]) to solve the problem of drug diffusion through a thin membrane. These techniques are introduced after applying a one-parameter group transformation to Equation (1), classified as a partial differential equation with two independent variables, which was reduced by one and transformed into an ordinary differential equation. We accomplished our computations by designing the MATLAB code for each approach. The most important aim of our article is to know the validity, efficiency, and accuracy of the developed techniques by comparing the computed results with earlier numerical and analytical solutions [4,18]. To examine the convergence and accuracy of the developed methods, we compute error as the following:

$$L_\infty \ Error = \max_{1 \le i \le N} |C_{numerical}(x_i, t_1) - C_{exact}(x_i, t_1)| \tag{31}$$

where $L_\infty Error$ expresses the maximum error norm. *max* is the maximum value of the absolute difference between the numerical and exact drug concentration results in the interval [1, N].

Now, we begin to demonstrate the obtained results in order to determine the stability, convergence, and validity of DQM based on three different types of shape functions in the following.

Table 1 explains the effect of using uniform and non-uniform grid points (N) on calculating C(x,t) via PDQM for both cases at different times (t), with $x = 0.5, \mu = \frac{10^{-6}}{4}$, $\alpha = 2$, $Q = 81.95$, and $q = 1.16$. For uniform grid points, the results match with previous studies [4,18] at $N \ge 9$ and execution time 0.25 s, and at $N \ge 7$ and execution time 0.013 s for non-uniform grid points. Thus, using non-uniform grid points is the best way to avoid Runge's phenomenon. A non-uniform distribution (Gauss–Chebyshev–Lobatto discretization) is used as the following:

$$x_i = \frac{1}{2}\left[1 - \cos(\frac{i-1}{N-1}\pi)\right], \ (i = 1, 2, \dots N) \tag{32}$$

**Table 1.** Computation of $C(x,t)$ via PDQM for cases (1 and 2) at grid points (N) and time (t) for $x = 0.5$, $\mu = \frac{10^{-6}}{4}$, $\alpha = 2$, $Q = 81.95$, and $q = 1.16$.

| Uniform (N) | C(x, 25) | | C(x, 50) | | Non-Uniform (N) | C(x, 25) | | C(x, 50) | |
|---|---|---|---|---|---|---|---|---|---|
| | Case 1 | Case 2 | Case 1 | Case 2 | | Case 1 | Case 2 | Case 1 | Case 2 |
| 4 | 0.00235 | 0.00241 | 0.00325 | 0.00344 | 4 | 0.00167 | 0.00175 | 0.00259 | 0.00262 |
| 5 | 0.00201 | 0.00206 | 0.00211 | 0.00235 | 5 | 0.00087 | 0.00088 | 0.00237 | 0.00239 |
| 6 | 0.00138 | 0.00139 | 0.00198 | 0.00200 | 6 | 0.00071 | 0.00072 | 0.00140 | 0.00145 |
| 7 | 0.00099 | 0.00105 | 0.00145 | 0.00148 | 7 | 0.00069 | 0.00071 | 0.00138 | 0.00142 |
| 8 | 0.00072 | 0.00075 | 0.00140 | 0.00141 | 8 | 0.00069 | 0.00071 | 0.00138 | 0.00142 |
| 9 | 0.00069 | 0.00071 | 0.00138 | 0.00142 | 9 | 0.00069 | 0.00071 | 0.00138 | 0.00142 |
| 10 | 0.00069 | 0.00071 | 0.00138 | 0.00142 | 10 | 0.00069 | 0.00071 | 0.00138 | 0.00142 |
| 11 | 0.00069 | 0.00071 | 0.00138 | 0.00142 | 11 | 0.00069 | 0.00071 | 0.00138 | 0.00142 |
| Previous Studies [4,18] | 0.00069 | 0.00071 | 0.00138 | 0.00142 | | 0.00069 | 0.00071 | 0.00138 | 0.00142 |
| Execution time | 0.25 (second)—uniform $N \ge 9$ | | | | | 0.013 (second)—non-uniform $N \ge 7$ | | | |

Table 2 compares the PDQM to previous studies [4,18] in terms of calculating the $U(x)$, $C(x,t)$, and $L_\infty$ error norms in [0, 1] for both cases at distances (x) for a non-uniform grid $(N = 7)$, $t = 20$ min, $\mu = \frac{10^{-6}}{4}$, $\alpha = 2$, $Q = 81.95$, and $q = 1.16$. It is noticed that the $L_\infty$ error reaches $10^{-7}$, which ensures the validity of the PDQM at t = 20 min and different distances (x). In addition, the results match with previous works [4,18] in both cases with an error of about $10^{-6}$ and an execution time of 0.013 s for computing $U(x)$ and $C(x,t)$. All

of this proves that the PDQM is an efficient and effective method for solving the problem of drug diffusion through membranes.

**Table 2.** Computation of $U(x), C(x,t)$, and $L_\infty$ error norms in $[0, 1]$ via the PDQM for cases (1 and 2) at distances (x) for a non-uniform grid $(N = 7), t = 20$ min, $\mu = \frac{10^{-6}}{4}, \alpha = 2, Q = 81.95$, and $q = 1.16$.

| | PDQM | | | | Previous Studies [4,18] | | |
| | $U(x)$ | | $C(x, 20)$ | | $C(x, 20)$ | | $L_\infty$ |
| $x$ | Case 1 | Case 2 | Case 1 | Case 2 | Case 1 | Case 2 | |
|---|---|---|---|---|---|---|---|
| 0 | 163.9000 | 163.9000 | 0.0008195 | 0.0008195 | 0.000820 | 0.000820 | $5.0 \times 10^{-07}$ |
| 0.1 | 151.9837 | 151.5689 | 0.0007599 | 0.0007584 | 0.000760 | 0.000758 | $1.0 \times 10^{-07}$ |
| 0.2 | 140.5881 | 140.5502 | 0.00070294 | 0.0007028 | 0.000703 | 0.000703 | $1.0 \times 10^{-07}$ |
| 0.3 | 129.8072 | 130.5885 | 0.000649 | 0.0006529 | 0.000650 | 0.000653 | $1.0 \times 10^{-06}$ |
| 0.4 | 119.7429 | 121.4970 | 0.0005987 | 0.0006075 | 0.000600 | 0.000608 | $1.3 \times 10^{-06}$ |
| 0.5 | 110.5066 | 113.1459 | 0.0005525 | 0.0005657 | 0.000553 | 0.000566 | $5.0 \times 10^{-07}$ |
| 0.6 | 102.2230 | 105.4577 | 0.00051112 | 0.000528 | 0.000512 | 0.000528 | $8.8 \times 10^{-07}$ |
| 0.7 | 95.0357 | 98.4086 | 0.00047518 | 0.000492 | 0.000476 | 0.000492 | $8.2 \times 10^{-07}$ |
| 0.8 | 89.11420 | 92.0381 | 0.00044447 | 0.0004602 | 0.000446 | 0.000461 | $1.5 \times 10^{-06}$ |
| 0.9 | 84.66554 | 86.47 | 0.00042333 | 0.0004325 | 0.000424 | 0.000433 | $6.7 \times 10^{-07}$ |
| 1 | 81.9500 | 81.9500 | 0.00040975 | 0.00040975 | 0.000410 | 0.000410 | $2.5 \times 10^{-07}$ |
| CPU (second) | | | | 0.013 (second) | | | |

Table 3 investigates the effect of some values on the accuracy of DSCDQM–RSK and DSCDQM–DLK, such as the band width (2M+1), the regularized Shannon factor (σ), the computational parameter (τ), and step size (Δ). Table 3 explains that DSCDQM–RSK is more accurate than DSCDQM–DLK in computing the concentration C(x,t) at t = 20 min compared with earlier solutions [4,18]. In addition, the bandwidth (2M+1 = 5) and $[\sigma = 1.45\Delta]$ are the most suitable choices for numerical results, which achieve more efficient results. It is noticed that the $L_\infty$ error reaches $10^{-7}$ and $10^{-8}$, which ensures the validity of DSCDQM–DLK and DSCDQM–RSK at t = 20 min and different distances (x), respectively. In addition, the results match with the previous literature [4,18] in both cases, with an error of about $10^{-8}$ and an execution time of 0.0108 s for computing C(x,t). All of this proves that DSCDQM–RSK is an efficient and effective method for analyzing drug diffusion through a thin membrane compared with previous methods, the PDQM, and DSCDQM–DLK.

The relative difference between two cases can be computed as follows:

$$\begin{aligned} R.D.\% &= \frac{U(x)_{case2} - U(x)_{case1}}{U(x)_{case2}} \times 100\% \\ &= \frac{C(x,t)_{case2} - C(x,t)_{case1}}{C(x,t)_{case2}} \times 100\% \end{aligned} \tag{33}$$

Thus, Table 4 shows the computation of $C(x,t)$ and $R.D.\%$ via DSCDQM–DLK and DSCDQM–RSK with band widths of $[2M + 1 = 5]$ and $[\sigma = 1.45\,\Delta]$ for both cases at distances (x) for $t = 100$ min, $\mu = \frac{10^{-6}}{4}, \alpha = 2, Q = 81.95$, and $q = 1.16$. The results demonstrated that the DSCDQM–RSK technique is the best; thus, it is used with a band width of [2M+1 = 5] and $\sigma[= 1.45\Delta]$ to present a parametric study for a drug diffusion problem through a thin membrane.

**Table 3.** Computation of $C(x,t)$ and $L_\infty$ error norms in [0, 1] via DSCDQM (DLK and RSK) with different band widths $[2M+1]$ for case (1) at distances (x) for $t = 20$ min, $\mu = \frac{10^{-6}}{4}$, $\alpha = 2$, $Q = 81.95$, and $q = 1.16$.

| x | 2M+1 | C(x,20) | | | | | | Previous Studies [4,18] | L∞ | |
|---|---|---|---|---|---|---|---|---|---|---|
| | | DSCDQM–DLK | DSCDQM–RSK | | | | | | DLK | RSK |
| | | | $\sigma = 1.2\Delta$ | $\sigma = 1.35\Delta$ | $\sigma = 1.45\Delta$ | $\sigma = 1.5\Delta$ | | | | |
| 0.2 | 3 | 0.000813 | 0.000787 | 0.000763 | 0.000752 | 0.000726 | | 0.000703 | $1 \times 10^{-7}$ | $1 \times 10^{-8}$ |
| | 4 | 0.0007072 | 0.000729 | 0.000718 | 0.000711 | 0.000701 | | | | |
| | 5 | 0.0007031 | 0.000718 | 0.000709 | 0.000703 | 0.000698 | | | | |
| | 6 | 0.0007031 | 0.000718 | 0.000709 | 0.000703 | 0.000698 | | | | |
| 0.4 | 3 | 0.0006274 | 0.000641 | 0.000635 | 0.000622 | 0.000629 | | 0.000600 | $1 \times 10^{-7}$ | $1 \times 10^{-8}$ |
| | 4 | 0.0006050 | 0.000631 | 0.000618 | 0.000608 | 0.000613 | | | | |
| | 5 | 0.0005999 | 0.000617 | 0.000608 | 0.000600 | 0.000595 | | | | |
| | 6 | 0.0005999 | 0.000617 | 0.000608 | 0.000600 | 0.000595 | | | | |
| 0.6 | 3 | 0.0005333 | 0.000541 | 0.000532 | 0.000528 | 0.000525 | | 0.000512 | $6 \times 10^{-7}$ | $1 \times 10^{-8}$ |
| | 4 | 0.0005222 | 0.000537 | 0.000522 | 0.000517 | 0.000516 | | | | |
| | 5 | 0.0005114 | 0.000530 | 0.000518 | 0.000512 | 0.000507 | | | | |
| | 6 | 0.0005114 | 0.000530 | 0.000518 | 0.000512 | 0.000507 | | | | |
| 0.8 | 3 | 0.0004529 | 0.000499 | 0.000472 | 0.000458 | 0.000449 | | 0.000446 | $3 \times 10^{-7}$ | $1 \times 10^{-8}$ |
| | 4 | 0.0004480 | 0.000480 | 0.000459 | 0.000450 | 0.000445 | | | | |
| | 5 | 0.0004457 | 0.000471 | 0.000452 | 0.000446 | 0.000441 | | | | |
| | 6 | 0.0004457 | 0.000471 | 0.000452 | 0.000446 | 0.000441 | | | | |
| CPU (second) | | 0.0115 (seconds) | | | | 0.0108 (seconds) | | | | |

**Table 4.** Computation of $C(x,t)$ and $R.D.\%$ via DSCDQM–DLK and DSCDQM–RSK with a band width of $[2M+1 = 5]$ and $[\sigma = 1.45\Delta]$ for cases (1 and 2) at distances (x) for $t = 100$ min, $\mu = \frac{10^{-6}}{4}$, $\alpha = 2$, $Q = 81.95$, and $q = 1.16$.

| x | DSCDQM–DLK | | | DSCDQM–RSK | | | Previous Studies [4,18] | | |
|---|---|---|---|---|---|---|---|---|---|
| | C(x,100) | | R.D.% | C(x,100) | | R.D.% | C(x,100) | | R.D.% |
| | Case 1 | Case 2 | | Case 1 | Case 2 | | Case 1 | Case 2 | |
| 0 | 0.004099 | 0.004099 | 0 | 0.00410 | 0.00410 | 0 | 0.00410 | 0.00410 | 0 |
| 0.1 | 0.003801 | 0.00379 | −0.29 | 0.00380 | 0.00379 | −0.26 | 0.00380 | 0.00379 | −0.26 |
| 0.2 | 0.003516 | 0.003515 | 0 | 0.00352 | 0.00352 | 0 | 0.00352 | 0.00352 | 0 |
| 0.3 | 0.003246 | 0.003266 | 0.61 | 0.00325 | 0.00327 | 0.61 | 0.00325 | 0.00327 | 0.61 |
| 0.4 | 0.002994 | 0.003038 | 1.44 | 0.00300 | 0.00304 | 1.315 | 0.00300 | 0.00304 | 1.315 |
| 0.5 | 0.002763 | 0.002829 | 2.33 | 0.00276 | 0.00283 | 2.33 | 0.00276 | 0.00283 | 2.33 |
| 0.6 | 0.002556 | 0.002637 | 3.07 | 0.00256 | 0.00264 | 3.03 | 0.00256 | 0.00264 | 3.03 |
| 0.7 | 0.002376 | 0.002461 | 3.45 | 0.00238 | 0.00246 | 3.25 | 0.00238 | 0.00246 | 3.25 |
| 0.8 | 0.002228 | 0.002302 | 3.21 | 0.00223 | 0.00230 | 3.18 | 0.00223 | 0.00230 | 3.18 |
| 0.9 | 0.002117 | 0.002162 | 2.08 | 0.00212 | 0.00216 | 2.08 | 0.00212 | 0.00216 | 2.08 |
| 1 | 0.002049 | 0.002049 | 0 | 0.00205 | 0.00205 | 0 | 0.00205 | 0.00205 | 0 |

Table 5 shows the results of C(x,t) for two cases at different distances (x) and times (t) using DSCDQM–RSK. The computed results demonstrated that C(x,t) reduces by a small amount with distance and increases with time. Tables 4 and 5 explain the tiny relative differences in C(x,t) for the two cases; therefore, the influence of the coefficient of diffusion in Equation (1) is negligible.

Figure 1 $U(x)$ is inversely proportional to specific sections of membrane (x), and the obtained results with the proposed schemes match with previous studies [4,18], which proves the validity, efficiency, and accuracy of the developed techniques.

At different times, Figures 2 and 3 show the variance of concentration with different sections of membrane (x) using DSCDQM–RSK for two cases. These figures demonstrate that the value of concentration increases with increasing time, but decreases with increasing membrane section (x).

**Table 5.** Computation of $C(x, t)$ and $R.D.\%$ via DSCDQM–RSK with [$2M+1 = 5$] and [$\sigma = 1.45\Delta$] for cases (1 and 2) at distances (x) and time (t) for $\mu = \frac{10^{-6}}{4}, \alpha = 2, Q = 81.95,$ and $q = 1.16$.

| x | $C(x, 25)$ | | | $C(x, 50)$ | | | $C(x, 75)$ | | |
|---|---|---|---|---|---|---|---|---|---|
| | Case 1 | Case 2 | R.D.% | Case 1 | Case 2 | R.D.% | Case 1 | Case 2 | R.D.% |
| 0 | 0.00102 | 0.00102 | 0 | 0.00205 | 0.00205 | 0 | 0.00307 | 0.00307 | 0 |
| 0.1 | 0.00095 | 0.00095 | 0 | 0.0019 | 0.0019 | 0 | 0.00285 | 0.00284 | −0.35 |
| 0.2 | 0.00088 | 0.00088 | 0 | 0.00176 | 0.00176 | 0 | 0.002637 | 0.00264 | 0.11 |
| 0.3 | 0.00081 | 0.00082 | 1.22 | 0.001623 | 0.00163 | 0.43 | 0.002435 | 0.00245 | 0.61 |
| 0.4 | 0.00075 | 0.00076 | 1.32 | 0.00150 | 0.00152 | 1.32 | 0.002246 | 0.00228 | 1.49 |
| 0.5 | 0.00069 | 0.00071 | 2.82 | 0.00138 | 0.00142 | 2.82 | 0.002073 | 0.00212 | 2.22 |
| 0.6 | 0.00064 | 0.00066 | 3.03 | 0.00128 | 0.00132 | 3.03 | 0.001918 | 0.00198 | 3.13 |
| 0.7 | 0.00059 | 0.00062 | 4.84 | 0.00119 | 0.00123 | 3.25 | 0.001783 | 0.00185 | 3.62 |
| 0.8 | 0.00056 | 0.00058 | 3.45 | 0.00111 | 0.00115 | 3.48 | 0.001672 | 0.00173 | 3.35 |
| 0.9 | 0.00053 | 0.00054 | 1.85 | 0.00106 | 0.00108 | 1.85 | 0.001588 | 0.00162 | 1.98 |
| 1 | 0.00051 | 0.00051 | 0 | 0.00102 | 0.00102 | 0 | 0.001537 | 0.001537 | 0 |

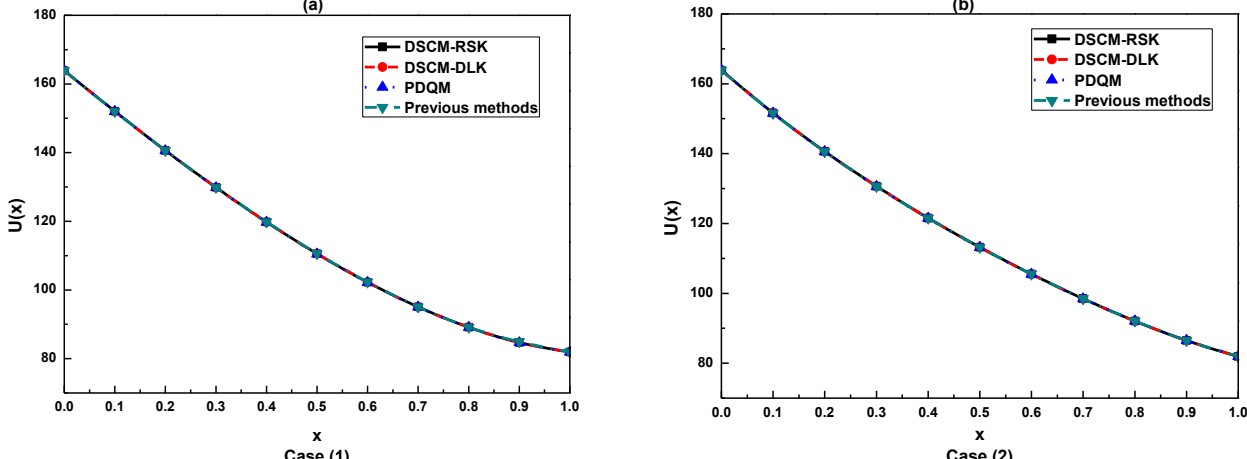

**Figure 1.** Variance of $U(x)$ with specific sections of membrane (x) via different methods for two cases (**a**) $P(x, t) = \frac{exp(-x)}{t}$ and (**b**) $P(x, t) = \frac{exp(-x^2)}{t}$, where $t = 100$ min, $2M + 1 = 5, \sigma = 1.45\Delta, \mu = \frac{10^{-4}}{4}, Q = 81.95,$ and $q = 1.16$.

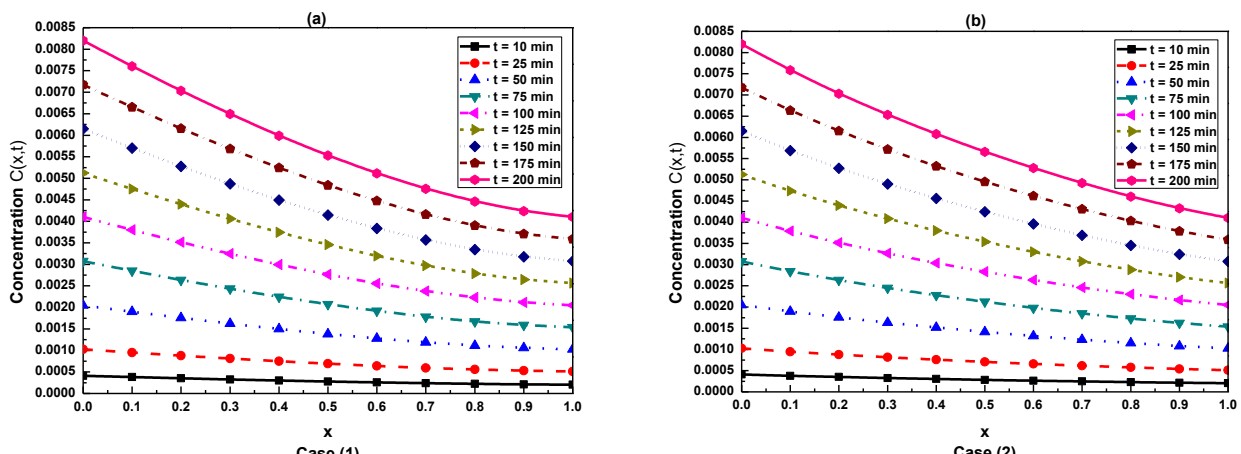

**Figure 2.** Variance of concentration $C(x, t)$ with specific sections of membrane (x) via DSCDQM–RSK for two cases (**a**) $P(x, t) = \frac{exp(-x)}{t}$ and (**b**) $P(x, t) = \frac{exp(-x^2)}{t}$ at different times (t), where $2M + 1 = 5, \sigma = 1.45\Delta, \mu = \frac{10^{-4}}{4}, Q = 81.95,$ and $q = 1.16$.

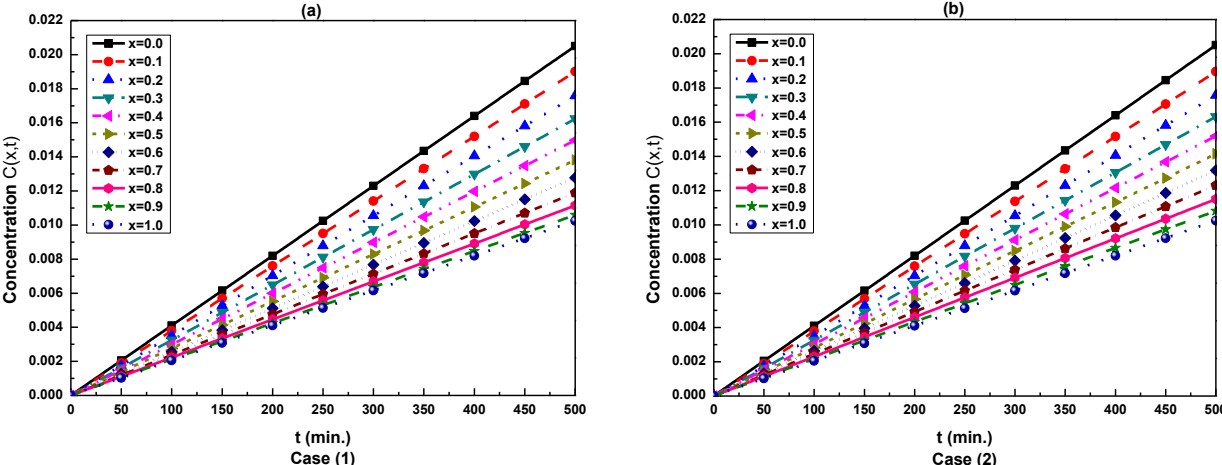

**Figure 3.** Variance of concentration $C(x,t)$ with time (t) via DSCDQM–RSK for two cases (**a**) $P(x,t) = \frac{exp(-x)}{t}$ and (**b**) $P(x,t) = \frac{exp(-x^2)}{t}$ at different specific sections of membrane (x), where $2M + 1 = 5, \sigma = 1.45\Delta, \mu = \frac{10^{-4}}{4}, Q = 81.95,$ and $q = 1.16$.

Figure 4 shows that the value of R.D.% grows gradually toward x, reaching its highest value at x = 0.7, before decreasing to zero at x = 1.

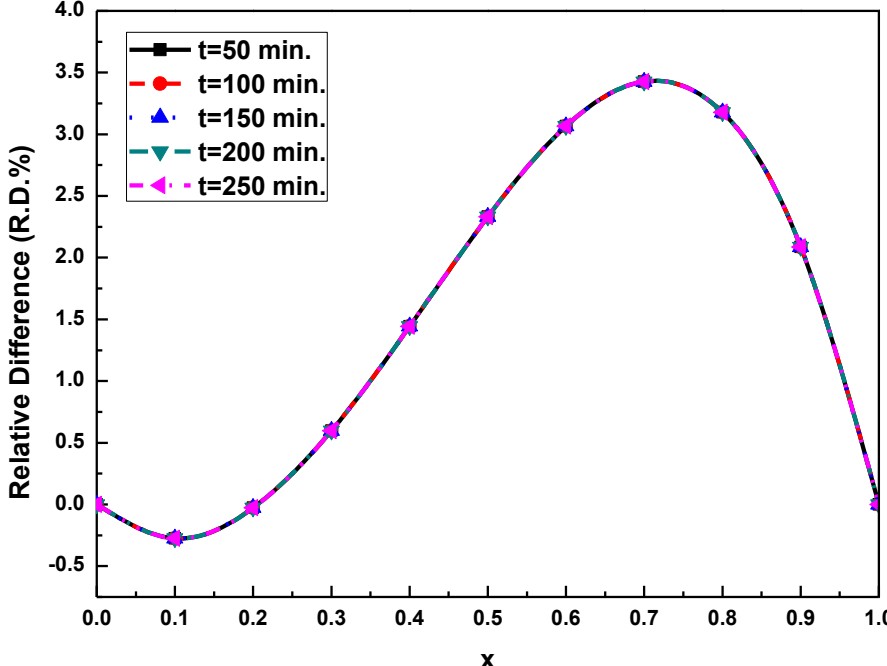

**Figure 4.** Variance of relative difference ($R.D.\%$) with specific sections of membrane (x) via DSCDQM–RSK for two cases at different times (t), where $2M + 1 = 5$, $\sigma = 1.45\Delta$, $\mu = \frac{10^{-4}}{4}, Q = 81.95$, and $q = 1.16$.

Figure 5 depicts the variation in donor D(t) and recipient R(t) cell concentrations over time (t). The concentration in the donor cell D(t) decreases with time, while that in the recipient cell R(t) rises, which is consistent with the theoretical model.

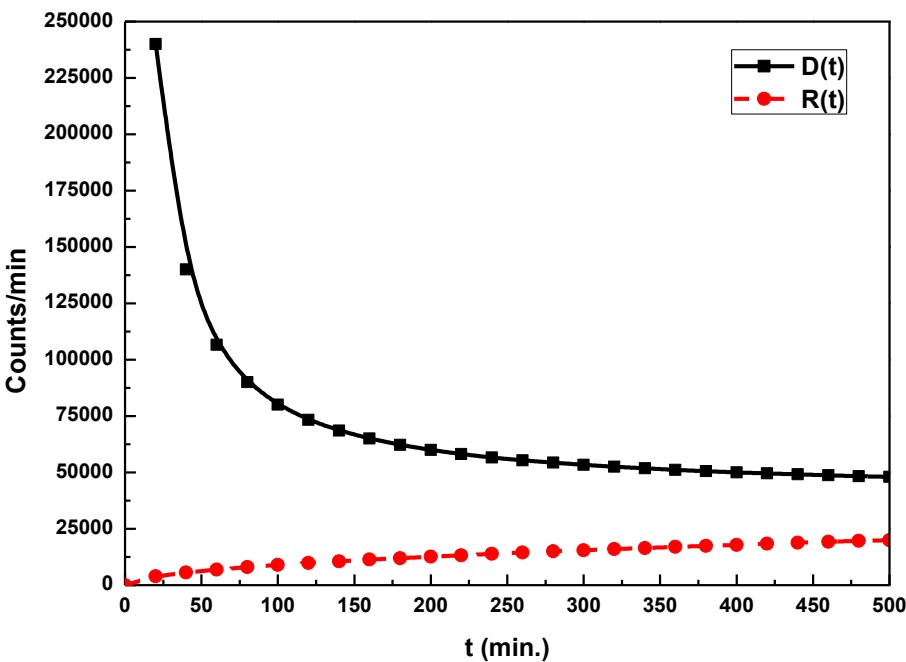

**Figure 5.** Variance of the donor $D(t)$ and receiver $R(t)$ cells concentrations with time (t) via DSCDQM–RSK, where $2M+1 = 5$, $\sigma = 1.45\Delta$, $\beta = 40,000$, $\mu = \frac{10^{-4}}{4}$, and $\gamma = 10^{-11}/32$.

## 5. Conclusions

Here, efficient numerical approaches are applied to solve the problem of drug diffusion through a thin membrane. The proposed technique is based on various shape functions such as Lagrange interpolation polynomials, delta Lagrange, and regularized Shannon kernels for the differential quadrature method (DQM). A nonlinear partial differential equation with two time- and space-dependent variables governs this problem. To decrease the two independent variables by one, a one-parameter group transformation is used and the partial differential equation is converted into an ordinary differential equation. Then, with the aid of the iterative technique, differential quadrature methods change this equation into an algebraic equation. Then, using a MATLAB program, a code that solves this equation for each shape function is created. To ensure the validity, efficiency, and accuracy of the developed techniques, the computed results are compared to previous numerical and analytical solutions.

We solved the problem using the techniques presented, and the numerical confirmation verified the previously established solutions [4,18]. We also confirmed the convergence of the offered techniques by computing L∞ error norms. As a result, our methods produce much more accurate, stable, and efficient results. A comparison between PDQM, DSCDQM–DLK, DSCDQM–RSK, and previous methods [4,18] in both cases at t = 100 min is presented. The results demonstrated that the DSCDQM–RSK technique is the best; thus, it is used to present a parametric study for a drug diffusion problem through a thin membrane. The best values for the parameters controlling our methods are a band width of [2M+1 = 5] and [σ = 1.45Δ], which are obtained with errors of $10^{-7}$ and $10^{-8}$ and an execution time = 0.0108 s. The novelty of this method is that it provides a parametric analysis in two cases to explain how the time and membrane section affect the drug concentration and diffusion coefficient for the first time as follows:

- The computed results demonstrated that the concentration reduces slightly with distance and increases with time.
- The tiny relative differences in concentration for the two cases prove that the influence of the coefficient of diffusion is negligible.
- The value of R.D.% grows gradually toward x, reaching its highest value at x = 0.7, before decreasing to zero at x = 1.

- The concentration in the donor cell D(t) decreases with time, while that in the recipient cell R(t) rises, which is consistent with the theoretical model.

Furthermore, it is demonstrated that the proposed techniques have a one-of-a-kind ability to solve such problems with initial and boundary conditions. As a result, we anticipate applying these techniques to other nonlinear partial differential problems in a variety of applied sciences.

**Author Contributions:** Conceptualization, M.M. and R.S.S.; methodology, M.M.; software, M.M.; validation, A.M. and R.S.S.; formal analysis, investigation, M.M.; resources, A.M.; data curation, writing—original draft preparation, M.M.; writing—review and editing, R.S.S.; visualization, supervision, A.M. All authors have read and agreed to the published version of the manuscript.

**Funding:** The Deanship of Scientific Research at the Islamic University of Madinah provided support to the Post-Publishing Program (2).

**Data Availability Statement:** The data presented in this study are available in the article.

**Acknowledgments:** The researchers wish to extend their sincere gratitude to the Deanship of Scientific Research at the Islamic University of Madinah for the support provided to the Post-Publishing Program (2).

**Conflicts of Interest:** The authors declare no conflict of interest.

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
