# Peer review of "Semi-Analytical Analysis of Drug Diffusion through a Thin Membrane Using the Differential Quadrature Method"

_mathematics, doi:10.3390/math11132998_

Round 1
Reviewer 1 Report
w
1. References are rather old. The following two review papers should be cited in the revised manuscript, one is for DQM and the other for DSC:
Tornabene, F., Fantuzzi, N., Ubertini, F., and Viola, E., 2015, “Strong Formulation Finite Element Method Based on Differential Quadrature: A Survey,” Appl. Mech. Rev., 67: p. 020801
X Wang, Z Yuan, J Deng. A Review on the Discrete Singular Convolution Algorithm and Its Applications in Structural Mechanics and Engineering. Archives of Computational Methods in Engineering, 2020, 27(5) 1633-1660
2. Some symbols have not been defined and their definition should be given. For example, D and R in Eq. (4), etc.
3. Eq. (2) is initial condition, but Eq. (3) seems not boundary conditions. Something (0<=x<=1) is wrong there? Should 0<=x<=1 be x=0 and x=1?
4. “So, the key to DQM accuracy is finding the weighting coefficients”----The distribution of grid points is another key for successfully using DQM. Therefore, the grid points should be given. For uniformly distributed grid points, N should be small to avoid Runge’s phenomenon. The information of N should be also provided. For DSC, why did authors use 2M+1=5? Sigma depends on M.
5. Reference format should be consistent and follow the requirement of the journal.
none
Author Response
Please find the attached file, and we are pleased for your valuable comments

Reviewer 2 Report
The article fits the theme and is well organized. However, there are a number of questions:
1. There are some strange signs in formula (31). Explain what the first sign in the right part of (31) means.
2. You write lines 201-203: "The most important aim of our article is to know the validity, efficiency, and accuracy of the developed techniques by comparing the computed results with earlier numerical and analytical solutions". But the choice of shape functions affects the accuracy of the method. Is it correct to compare the accuracy of methods with completely different shape functions? You need to describe in detail how you chose these functions.
3. When comparing PDQM, DSCDQM-DLK, and DSCDQM-RSK methods, the authors refer to Figure 1. They conclude that the DSCDQM-RSK method is the best. However, all of the curves in Figure 1 have merged into a single line. This figure does not demonstrate the advantage of any one method.
4 The authors write (lines 269-270): "Figure 4 shows that the value of R.D. % grows gradually toward x, reaching its highest value at x = 0.7, before decreasing to zero at x = 1. This can be clearly seen in Figure 4. Explain why this is happening, however. What is the reason for this behavior? What are the reasons?
Once corrected, the paper can be published.
Author Response
The authors are grateful to the reviewer for his corrections that have been used to improve the quality of the manuscript. The comments below have also been used to update the manuscript and we are grateful to the editor and reviewer for their meaningful contributions.
Please find the attached file

Reviewer 3 Report
Comments on “Semi-Analytical Analysis for Drug Diffusion through a Thin Membrane Using the Differential Quadrature Method”
§ What specific problem or challenge is being addressed in this research article?
§ Can you provide a more detailed explanation of the differential quadrature approach and how it is applied to solve the problem of drug diffusion through a thin membrane?
§ What are the reasons for choosing Lagrange interpolation and discrete singular convolution as the shape functions for the differential quadrature method?
§ How is the nonlinear partial differential equation with two time- and space-dependent variables derived? Can you provide more insight into the governing equation?
§ What is the rationale behind transforming the partial differential equation into an ordinary differential equation using a one-parameter group transformation? How does this reduction help in the analysis?
§ Could you explain the iterative technique used to convert the differential equation into an algebraic equation in more detail?
§ What is the significance of using a MATLAB program to implement the differential quadrature method? Are there any specific advantages or limitations associated with this choice?
§ How are the computed results compared to previous numerical and analytical solutions? Can you explain the specific metrics and techniques used for comparison?
§ What is the L∞ error and how is it applied to assess the validity, efficiency, and accuracy of the developed techniques? Are there any other error measures considered?
§ Can you provide a summary of the main findings and conclusions drawn from the numerical outcomes of the study?
§ In what specific aspects does the differential quadrature method, based on discrete singular convolution shape function, outperform other techniques? Are there any limitations or areas where it may not be as effective?
§ Were there any challenges or difficulties encountered during the implementation of the MATLAB program? How were they addressed?
§ Are there any assumptions made during the derivation or implementation of the differential quadrature method that could potentially affect the accuracy or applicability of the results?
§ Is there any sensitivity analysis conducted to evaluate the robustness of the developed techniques? If so, what parameters or variables were investigated?
§ Are there any potential applications or extensions of the proposed approach beyond the specific problem of drug diffusion through a thin membrane?
§ How generalizable are the findings of this study? Can they be applied to different types of membranes or drug diffusion scenarios?
§ Were there any limitations or constraints that may have affected the scope or implementation of the research? If so, what steps were taken to mitigate their impact?
§ Can you provide more details on the computational efficiency and convergence rate of the differential quadrature method compared to other techniques? Were any specific benchmarks or performance metrics used?
§ What are the main contributions of this research article to the existing body of knowledge in the field? How does it advance the understanding or application of differential quadrature methods?
§ Are there any specific recommendations or suggestions for future research based on the findings and limitations of this study?
§ How robust is the reliability of the developed techniques, considering potential variations or uncertainties in the system parameters or boundary conditions?
§ Were there any ethical or regulatory considerations involved in the research, particularly regarding the application of drug diffusion through a thin membrane? If so, how were they addressed?
§ Are there any additional experiments or analyses that could further validate or strengthen the claims made in the article?
Moderate editing of English language required.
Author Response

(The authors gave the same response as above.)

Round 2
Reviewer 2 Report
The authors responded sufficiently to all the comments. I believe that the article can be published as presented.
Reviewer 3 Report
I'm pleased to inform you that you have effectively addressed all the queries raised by me. After reviewing your responses, I must acknowledge that the manuscript has significantly improved. Your efforts and thoroughness in addressing the concerns are commendable.